TECHNICAL RELEASE

# BatchEval Pipeline: batch effect evaluation workflow for multiple datasets joint analysis

Chao Zhang[1], Qiang Kang[1], Mei Li[1], Hongqing Xie[1], Shuangsang Fang[1,2] and Xun Xu[3,*]

1  BGI Research, Shenzhen, 518103, China
2  BGI Research, Beijing, 102601, China
3  BGI Research, Wuhan, 430074, China

## ABSTRACT

As genomic sequencing technology continues to advance, it becomes increasingly important to perform joint analyses of multiple datasets of transcriptomics. However, batch effect presents challenges for dataset integration, such as sequencing data measured on different platforms, and datasets collected at different times. Here, we report the development of BatchEval Pipeline, a batch effect workflow used to evaluate batch effect on dataset integration. The BatchEval Pipeline generates a comprehensive report, which consists of a series of HTML pages for assessment findings, including a main page, a raw dataset evaluation page, and several built-in methods evaluation pages. The main page exhibits basic information of the integrated datasets, a comprehensive score of batch effect, and the most recommended method for removing batch effect from the current datasets. The remaining pages exhibit evaluation details for the raw dataset, and evaluation results from the built-in batch effect removal methods after removing batch effect. This comprehensive report enables researchers to accurately identify and remove batch effects, resulting in more reliable and meaningful biological insights from integrated datasets. In summary, the BatchEval Pipeline represents a significant advancement in batch effect evaluation, and is a valuable tool to improve the accuracy and reliability of the experimental results.

**Availability & Implementation:** The source code of the BatchEval Pipeline is available at https://github.com/STOmics/BatchEval.

**Subjects**  Imaging, Bioinformatics, Cell Biology

**Submitted:**  12 June 2023

\* Corresponding author. E-mail: xuxun@genomics.cn

Preprint submitted at https://doi.org/10.1101/2023.10.08.561465

Included in the series: *Spatial Omics: Methods and Application* (https://doi.org/10.46471/GIGABYTE_SERIES_0005)

## INTRODUCTION

Advancements in gene sequencing technology have facilitated the integrated analysis of multiple batches of gene transcription data, resulting in more reliable information extracted from datasets. However, batch effects arising from differences in sequencing platforms, experimental designs, experimentalists, laboratory conditions, and experimental reagent batches are often neglected during joint analysis. Batch effects introduce technical biases into sequencing and reduce the dependability of downstream analysis. Several efficient approaches have been proposed to minimize technical biases and batch effects in integrated datasets, while retaining the most significant biological variance. These approaches include non-linear models, such as Seurat's classic correlation analysis [1], linear regression models, such as Combat [2], and implementations based on matching mutual nearest neighbors, such as MNNs [3]. In some differential gene expression analysis

models, such as MSAT [4], DESeq [5] and Limma [6], batch effect is integrated into the model as a significant component to eliminate it from the dataset. More recently, works have been published to improve the integration of spatially resolved transcriptomics by exploiting spatial information, such as spatiAlign [7] and PRECAST [8].

Visualization tools are useful for the initial assessment of dataset integration, and to rapidly compare the results after removing batch effect. Density distribution visualizations show differences in gene expression counts between different tissue sections. Overlays of probability density functions compare variations across sections. Quantitative statistics, such as the chi-square test [9] and the local inverse Simpson index (*LISI*) [10], precisely quantify dataset integration.

When integrating datasets from multiple/different tissue sections, it may not be clear whether batch effect exists, or which removal method should be used, thus requiring detailed analysis. Even though a user-friendly software such as BatchQC has been published [11], there is still a shortage of tools suitable for analyzing large-scale dataset integration.

To address these issues, we developed the BatchEval Pipeline, a comprehensive workflow for evaluating batch effect on large-scale dataset integration. The BatchEval Pipeline evaluates dataset integration from different perspectives related to data mixing and biological variance preservation after batch effect removal. The BatchEval Pipeline generates a report, which consists of a series of HTML pages, including a main page, a raw integration dataset evaluation page, and several pages of evaluation results for each built-in batch effect removal method. The main page exhibits details of integrating the datasets—a comprehensive score for evaluating batch effect on raw datasets, a summary of the results for batch effect removal using different built-in methods (including single-cell-based methods, such as Harmony [10] and BBKNN [12], and spatially resolved transcriptomics method, such as spatiAlign [7]), and the most recommended method to remove batch effect. Users can conveniently access the detailed report of raw integrated datasets and the batch effect removal method through a super link in the main page. The raw dataset page consists of a statistical evaluation, a biological variance preservation metric score, and several visualization panels. The biological variance preservation metric score and visualization panels may be assessed on the batch effect removal method pages as well. A top left button enables users to navigate back to the main page from any page in the workflow. The BatchEval Pipeline workflow is shown as in Figure 1.

The main contributions of this study are as follows:

(1)  The BatchEval Pipeline carefully analyzes batch effect from integrated datasets, considering various factors related to data mixing, and biological variance preservation before and after dataset integration.

(2)  The BatchEval Pipeline can provide extensible remove batch effect benchmarking and generates an evaluation report, which includes single-cell-based and spatially resolved transcriptomics methods. The BatchEval Pipeline can also recommend suitable batch effect removal methods according to different datasets.

(3)  Evaluation results of the BatchEval Pipeline are crucial in determining whether datasets need batch effect correction and provide solutions for such corrections.

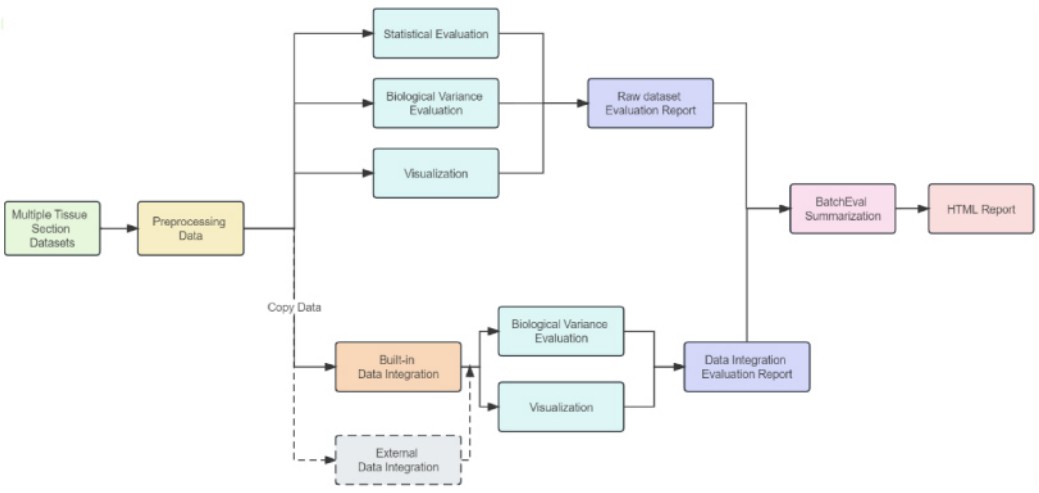

**Figure 1.** BatchEval Pipeline workflow.

## RESULTS

### Dataset

We collected two groups of spatially resolved transcriptomics datasets to evaluate the BatchEval Pipeline. (1) Two mouse brain olfactory bulb (OB) datasets were measured by Stereo-seq [13], and 10× Genomics Visium [14], respectively. Both OB datasets contained five identical cell type annotations. The Stereo-seq dataset included 1123 spots, each of which had an average of 6028 genes and an average gene expression of 16681. The 10× Genomics Visium dataset included 1184 spots, each of which had an average of 4580 genes and an average gene expression of 16916. (2) Five time-series mouse embryonic brain datasets were measured by Stereo-seq. These brain sections were collected from different embryonic tissues from E9.5 to E15.5, which included a total of 81181 spots/cells and 22864 genes in the merged dataset.

### Statistical analysis with BatchEval Pipeline

Batch effect and sequencing outcomes are typically complex and influenced by various factors, such as experimental conditions, operators, reagents, and timing, which are not directly related to the experiment's goal. However, batch effect can impact the accuracy of downstream results. A dataset includes a variety of batch effects, as well as associations between linear and non-linear processes, which can affect the distribution of gene expression probability density function for each spot. Additionally, different sequencing techniques can result in varying sequencing coverage depths across data batches. To address these issues, the BatchEval Pipeline performs Min–Max normalization and logarithmic mapping preprocessing on each spot/cell gene expression levels and integrates multiple batches of gene expression data into low-dimensional representations.

The BatchEval Pipeline employs Kruskal–Wallis H test [15] to evaluate the variation in the average level of gene expression across different tissue sections, and performs variance analysis on gene expression total counts for each tissue section. However, when comparing data from different platforms, such as Stereo-seq and 10× Genomics Visium, it is important to note that they may have considerable differences and not satisfy the homogeneity of

**Table 1.** Statistical evaluation results of mouse olfactory bulb dataset before data integration.

| Variation analysis | *n_batch* | *n_sample* | *F* | *p* value | *F* ref (2, 3116) |
|---|---|---|---|---|---|
| | 3 | 3119 | 252.6876 | 0 | 2.9986 |
| K–S test | *n_sample* | | *k–s stat* | | *p* value |
| batch0–1 | 1935 | | 0.6924 | | 0 |
| batch0–2 | 1996 | | 0.6115 | | 0 |
| batch1–2 | 2307 | | 0.1579 | | 0 |
| Cramer's test | Pearson correlation coefficient | | Cramer's V coefficient | | |
| | 0.9270 | | 0.8190 | | |

**Table 2.** Statistical evaluation results of mouse embryonic brain datasets before data integration.

| Variation analysis | *n_batch* | *n_sample* | *F* | *p* value | *F* ref (2, 3116) |
|---|---|---|---|---|---|
| | 5 | 81181 | 20433.7507 | 0 | 2.3720 |
| K–S test | *n_sample* | | *k–s stat* | | *p* value |
| batch0–1 | 16673 | | 0.6706 | | 0 |
| batch0–2 | 20986 | | 0.5892 | | 0 |
| batch0–3 | 24791 | | 0.6569 | | 0 |
| batch0–4 | 28064 | | 0.7920 | | 0 |
| batch1–2 | 31437 | | 0.3494 | | 0 |
| batch1–3 | 35242 | | 0.2209 | | 0 |
| batch1–4 | 38515 | | 0.6507 | | 0 |
| batch2–3 | 39555 | | 0.1641 | | 0 |
| batch2–4 | 42828 | | 0.7101 | | 0 |
| batch3–4 | 46633 | | 0.7073 | | 0 |
| Cramer's test | Pearson correlation coefficient | | Cramer's V coeffient | | |
| | 0.9775 | | 0.9005 | | |

variance assumption. Therefore, it may not be clear how gene expression consistency differs between the data from two platforms. To determine if gene expression data from several batches originated from the same distribution, a Kolmogorov–Smirnov Test [16] was performed. By observation, the data from Stereo-seq and 10× Genomics Visium expression did not fit together (Table 1).

The BatchEval Pipeline utilizes a contingency table to analyze the correlation between experimental conditions and dataset batches. For example, Cramer's V correlation coefficient [17] in Table 1 was calculated based on it. By assessing the correlation between different experimental conditions for each batch, the pipeline determined that the experimental conditions in the Stereo-seq and 10× Genomics Visium dataset batches were closely related, with a Cramer's V correlation coefficient of approximately 0.8190 (the statistical evaluation results of mouse embryonic brain datasets are shown in Table 2).

## Biological variance preservation evaluation with BatchEval Pipeline

The BatchEval Pipeline adopts a non-linear neural network classifier to estimate data mixing across multiple tissue sections. The model takes cells/spots gene expression matrices as inputs and predicts tissue sections from which each cell/spot come from. If the prediction accuracy is low, it means that the integrated dataset is well mixed, otherwise, the integrated dataset is poorly mixed. As shown in Table 3, the batch/domain estimate score, accept rate of mouse embryonic brain datasets are around 2%, indicating that the classifier can successfully differentiate which spot/cell comes from which tissue sections. Therefore, it was concluded that there is a definite batch effect between each time-series mouse embryonic brain dataset.

**Table 3.** Biological variance evaluation results of mouse embryonic brain datasets before data integration.

| Batch/domain estimate score | *n_batch* | *n_sample* | Train size | Accept rate |
|---|---|---|---|---|
| | 5 | 81181 | 56826 | 0.0194 |
| k-BET score | chi mean | 95% *p* value | Accept rate | Reject rate |
| | 58.0453 | 0.0027 | 0.0085 | 0.9915 |
| Local inverse Simpson's index | *iLISI* | *cLISI* | F1 score |
| | 0.0154 | 0.0386 | 0.0303 |
| Silhouette score | *iSS* | *cSS* | F1 score |
| | 0.6675 | 0.6155 | 0.4318 |

Exploratory transcriptome downstream analysis, especially low-dimensional embedding techniques such as principal component analysis (PCA) and clustering, are commonly used in batch effect evaluation methods. These methods are used to assess the efficiency of dataset integration, which can be divided into two parts: retaining biological variances and testing the validity of dataset integration/data mixing. BatchEval Pipeline employs k-BET test [9] to evaluate the mixing level between different tissue sections in the neighborhood surrounding each spot/cell. This method first projects integrated datasets into a low-dimensional embedding by PCA or Uniform Manifold Approximation and Projection (UMAP) [18], then applies Pearson's Chi-square test. If the neighborhood of each spot/cell is from various tissue sections, the relevant data mixing level is more acceptable, indicating that the data are mixing well. In the low-dimensional embedding, the dataset can be clearly distinguished, and the batch effect is evident. As shown in Table 3, the distributions of time-series mouse embryonic brain datasets differ significantly, and the k-BET score accept rate is approximately 0.0085 ($p$ = 0.0027 < 0.05).

The BatchEval Pipeline presumes that cell type labels are accessible for the data, and utilizes the *LISI* [10] and silhouette coefficient (*SS*) to estimate the data mixing and biological variance preservation while removing batch effect. After integrating and projecting the data into a low-dimensional embedding, the Pipeline computes the *LISI* and *SS* using two different groupings: (1) grouping using different tissue sections as the batch *LISI* (*iLISI*)/batch *SS* (*iSS*) score, and (2) grouping known cell types or clustering types as the cell/domain-type *LISI* score (*cLISI*) or cell/domain-type SS score (*cSS*). The Pipeline then uses F1 score to summarize the *LISI* and SS score, respectively. A larger F1 score of *LISI*/*SS* suggests better dataset integration that preservers the biological variations between domain cluster types while removing batch effect across multiple tissues. As shown in Table 3, the F1 scores of *LISI* and *SS* were low, suggesting that the raw datasets of time-series mouse embryonic brain datasets were not well mixed. Thus, the dataset requires further batch effect removal processing (the statistical evaluation results of mouse olfactory bulb dataset are shown in Table 4).

## Visualization of BatchEval Pipeline

In Figure 2, we exhibit the statistical analysis results of mouse embryonic brain datasets before data integration. The kernel distribution curve of gene expression total counts (Figure 2a) and cumulative density curve (CDF) (Figure 2b) showed significant differences. We concluded that greater the maximum difference value between the CDF curve, the more significant the batch effect. The total counts of gene expression for each spot/cell also showed differences (Figure 2c). Furthermore, we evaluated the mean and variance of each gene (Figure 2d), and observed that "batch 4" genes were more discrete than in other tissues.



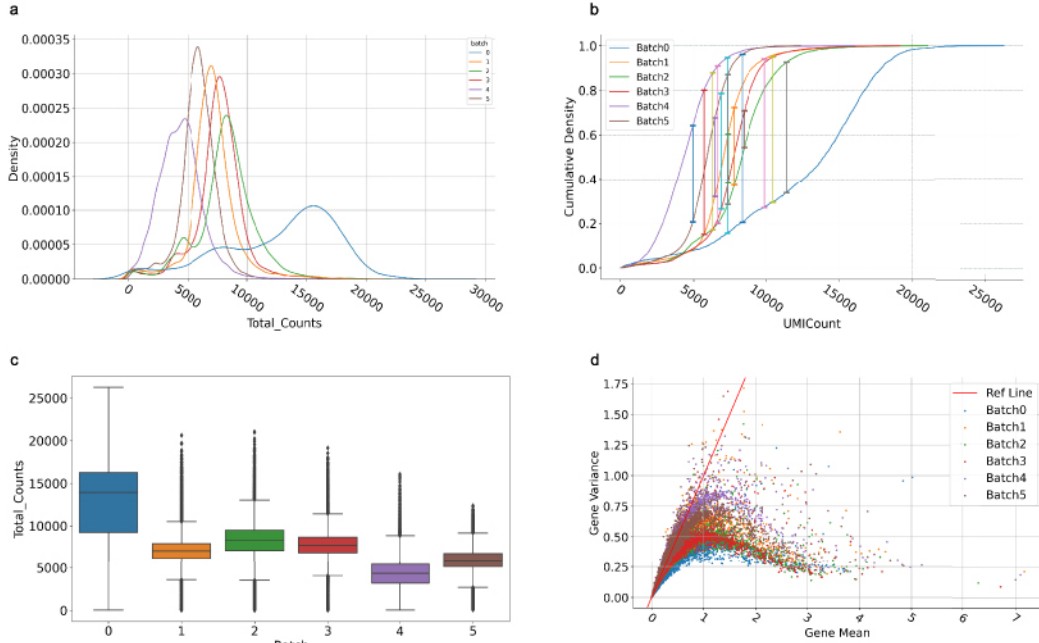

**Figure 2.** Visualization of statistical analysis for time-series mouse embryonic brain datasets. (a) Kernel distribution curve of total gene expression counts for each tissue section, different color represents different tissues. (b) Cumulative density curve of total gene expression counts for each tissue section. (c) Boxplot of total gene expression counts. (d) The scatter plot of mean gene and gene variance.

**Table 4.** Biological variance evaluation results of mouse olfactory bulb dataset before data integration.

| Batch/domain estimate score | *n_batch* | *n_sample* | Train size | Accept rate |
|---|---|---|---|---|
|  | 3 | 3119 | 2183 | 0 |
| k-BET score | chi mean | 95% *p* value | Accept rate | Reject rate |
|  | 29.8851 | 0 | 0 | 1 |
| Local inverse Simpson's index | *iLISI* | *cLISI* | F1 score |
|  | 0.0034 | 0.0888 | 0.0067 |
| Silhouette score | *iSS* | *cSS* | F1 score |
|  | 0.7842 | 0.4911 | 0.2999 |

Furthermore, the BatchEval Pipeline employs PCA to reduce the dimension of merged dataset. The scatter plots and edge probability density (PDF) curves illustrate the variations in distribution across multiple data batches in PCA dimensions. As shown in Figure 3a, the first two PCA dimensions were used as horizontal and vertical coordinates, and a PDF curve was fitted to capture the variations in the distribution of gene expression in different batches. The BatchEval Pipeline can quickly display the level of data mixing across multiple batches using UMAP and various color displays based on batch metadata. We utilized the integrated dataset in UMAP space and colored by "batch". As shown in Figures 3b, c and d, we observed that spatiAlign efficiently mixed time-series mouse embryonic brain datasets, however, other methods did not achieve the efficiency of spatiAlign.

Since the distribution of gene expression levels in different batches is usually unknown, we also employed quantile-quantile plots (Q–Q plot) to evaluate data distribution, which can be used to assess whether data from different batches follow the same distribution [19]. Data from multiple batches were coupled with each other to generate Q–Q maps for



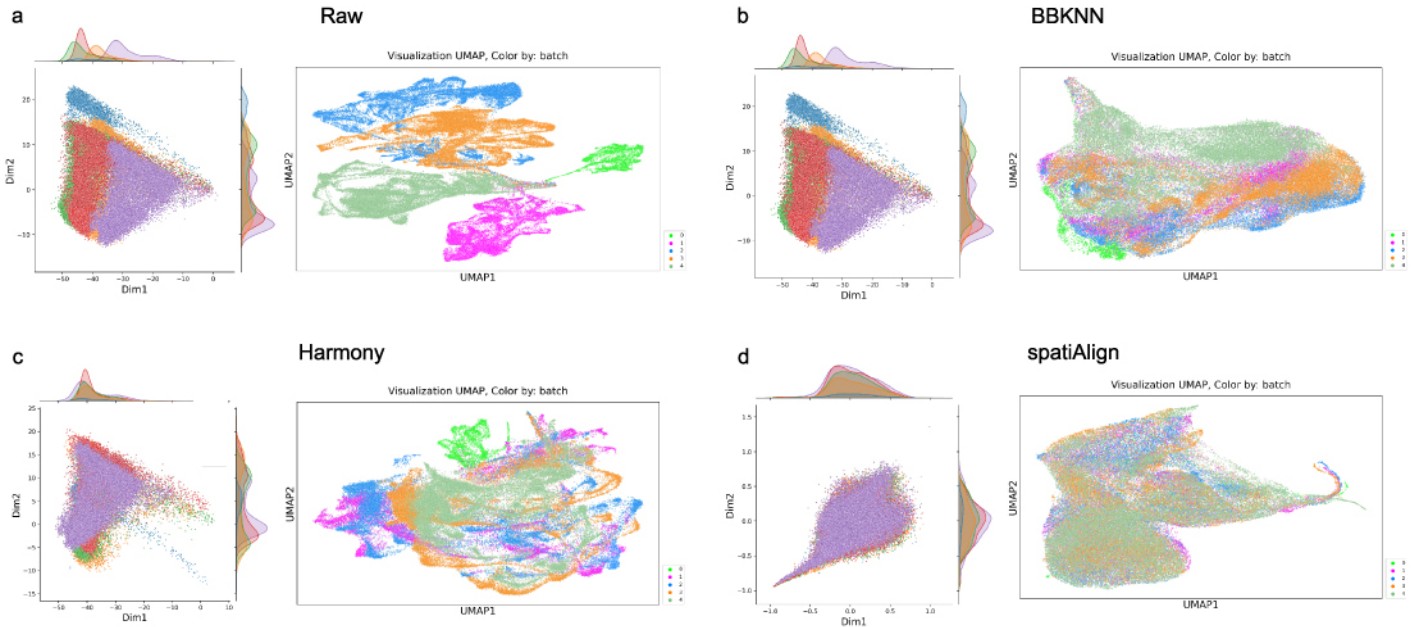

**Figure 3.** Visualization of integration for raw datasets and built-in batch effect removal methods. (a) Visualization of raw dataset. (b) Visualization of dataset integrated by BBKNN. (c) Visualization of dataset integrated by Harmony. (d) Visualization of dataset integrated by spatiAlign. All sub-figures left, joint plot of PCA, which using first two PC component to plot, and right, UMAP plot, colored by different data tissue sections.

**Table 5.** The summary evaluation of time-series mouse embryonic brain datasets.

|  | Raw | spatiAlign | Harmony | BBKNN |
|---|---|---|---|---|
| k-BET score | 0.0085 | 0.7419 | 0.1055 | 0.1598 |
| 95% *p* value | 0.0027 | 0.3183 | 0.0266 | 0.0478 |
| *BatchEval score* | 0.1605 | 0.6315 | 0.5064 | 0.3086 |
| Conclusion | This dataset has batch effect and requires further processing. Recommend to use "spatiAlign". | | | |
|  | More details of "spatiAlign" can be found in "https://github.com/STOmics/Spatialign.git". | | | |

gene expression. When gene expression in the matched data was less different, the cumulative density distribution of the corresponding gene expression was more tightly spaced.

## BatchEval score evaluation

We employed biological variance preservation score before and after batch effect removal. We employed data integration mixing score to summarize the batch effect of dataset integration. For example, as shown in Table 5, we utilized mouse embryonic brain datasets for test, and concluded that this dataset requires further processing to remove batch effect, and the most recommended method is "spatiAlign". This information is essential for researchers to understand the reliability of downstream analyses and to make informed decisions about the suitability of the dataset for their research goals (the summaries of the evaluation results for mouse olfactory bulb datasets are shown in Table 6).

## METHOD

The gene expression profiles are stored in a matrix with rows representing cells/spots and columns representing genes in the dataset. To ensure consistent and reliable results in

**Table 6.** The summary evaluation of mouse olfactory dataset.

|  | Raw | spatiAlign | Harmony | BBKNN |
|---|---|---|---|---|
| k-BET score | 0 | 0.6916 | 0.0901 | 0.1828 |
| 95% *p* value | 0 | 0.2870 | 0.0228 | 0.0546 |
| *BatchEval score* | 0.1022 | 0.7035 | 0.5256 | 0.3305 |
| Conclusion | This dataset has batch effect and requires further processing. Recommend to use "spatiAlign". | | | |
|  | More details of "spatiAlign" can be found in "https://github.com/STOmics/Spatialign.git". | | | |

subsequent analyses, the datasets are preprocessed using Min–Max normalization and log mapping, which standardize the values to the same range and transform them to a logarithmic scale. The preprocessing can be calculated using the follow parameterization:

$$\hat{x} = \log 1p \left( \frac{x - x_{\min}}{x_{\max} - x_{\min}} \right) \tag{1}$$

where *x* is the original expression of each spot/cell, $x_{\min}$ and $x_{\max}$ are the minimum and maximum expressions captured in the dataset, respectively.

## Statistical test

The Kruskal–Wallis H test [15] mixes the gene expression of each spot in different batches, sorts them from the smallest to largest, and records the ordinal number (rank). If the gene expression values in the sorted spot are the same, the corresponding rank is the same. Then, rank sum of each value is calculated with the Student's *T* test. The value of *H* indicates the distribution of rank in *k* batches. The larger the value of *H*, the greater the difference in rank. The calculation formula is shown as follows:

$$H = \frac{12}{N(N+1)} \sum_{i=1}^{k} \frac{R_i^2}{n_i} - 3(N+1) \tag{2}$$

where *N* denotes the number of all spots in different batches, *k* denotes the number of data batches, $n_i$ denotes the number of spots in batch *i*, and $R_i^2$ denotes the square of the rank sum of each spot in batch *i*.

Kolmogorov–Smirnov Test [16] can be used to test whether the cumulative density distributions of two datasets are different:

$$\text{stat} = \sup_x |F_1(x) - F_2(x)| \tag{3}$$

where *x* denotes the gene expression of spot and *F* (·) denotes cumulative density function.

Cramer's V correlation coefficient [17] can be used to determine correlation between several experimental conditions in each dataset. The first step would be to generate a cross-tabulation between each experimental condition and the batch. By default, the BatchEval Pipeline generates cross-tabulation using several experimental conditions for every batch. The calculation formula is as follows:

$$\phi_c = \sqrt{\frac{\chi^2}{N(k-1)}} \tag{4}$$

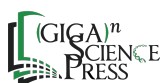

where $\chi^2$ denotes Pearson chi-square statistic, $N$ is the total number of samples tested in the cross-tabulation table, and $k$ number of categories of smaller variables in cross-tabulation table. Cramer's V correlation coefficient reflects the magnitude of correlation between data from different batches under different experimental conditions.

Neural network models can fit complex relationships between datasets. The BatchEval Pipeline constructs domain consistency assessment neural network models for estimating the magnitude of variation in different batches of datasets:

$$\hat{x} = \text{relu}\left(\sum_k w_k x_k + b\right) \tag{5}$$

where relu = max(0, $x$), $x$ denotes the gene vector of each spot, $w$ denotes the model weight, and $b$ denotes the model offset. When the number of spots included in each batch are inconsistent, BatchEval Pipeline employs Focal Loss [20] to measure the difference between model prediction results and labels, and the stochastic gradient descent method to update model parameters. The Focal Loss is calculated as follows:

$$\text{Focal\_Loss} = -\alpha(1-P)_t^\gamma \log(P_t) \tag{6}$$

where, $a$, $\gamma$ are hyperparameters and $P_t$ is predicted probability value of model output. The more accurate the results predicted by the model, higher the level of differentiability of the data domain in different batches and more obvious the batch effect.

## Metric score

The k-BET test [9] is based on the Pearson chi-square test to test the level of mixing of different batches of data within a local area and is calculated as follows:

$$\kappa_j^k = \sum_{i=1}^{l} \frac{(n_{ji}^k - f_i \times k)^2}{f_i \times k} \sim \chi_{l-1}^2 \tag{7}$$

where $\chi_{l-1}^2$ denotes the $\chi^2$ distribution with degree of freedom $l-1$ and $n_{ji}^k$ denotes the number of cells in subset $j$ of size $k$ in batch $i$. $f$ denotes the probability distribution of batch $i$ samples in the overall population.

To simultaneously evaluate the separation of each cell/batch cluster and the degree of mixing of multiple datasets, we calculated the average local inverse Simpson's index (*LISI*) [10] score of the datasets using two different groupings: (1) grouping using different datasets as the batch *LISI* score (*iLISI*), and (2) grouping known cell types or clustering types as the cell/domain-type *LISI* score (*cLISI*). In dataset integration, a larger value of *iLISI* indicated better mixing of datasets, and smaller *cLISI* indicated better preservation of the biological variance between cell/domain-types. These two metrics can be summarized using the F1 score as follows:

$$\text{F1 score}_{LISI} = \frac{2 \times (1 - cLISI) \times iLISI}{(1 - cLISI) + iLISI} \tag{8}$$

where $cLISI = \text{MinMaxNorm}(1/\sum_{b=1}^{B}p^2(b))$, $iLISI = \text{MinMaxNorm}(1/\sum_{c=1}^{C}p^2(c))$, $B$ and $C$ are the cell/domain-type or batch marker, $p$ ($b$) and $p$ ($c$) are the probability values of the cell/domain-type and batch in the local area, respectively.

Furthermore, we also implemented silhouette coefficient (*SS*) to evaluate the separation of each domain cluster and degree of mixing of multiple datasets. Similar to F1 score of *LISI*, we calculated the average *SS* of the datasets using two grouping labels, domain cluster type (*cSS*), different dataset type (*iSS*), respectively. These two metrics also can be summarized using F1 score as follows,

$$\text{F1 score}_{SS} = \frac{2 \times (1 - iSS') \times cSS'}{(1 - iSS') + cSS'} \tag{9}$$

where, $iSS' = (1 + iSS)/2$ and $cSS' = (1 + cSS)/2$. A larger F1 score of *SS* suggests better dataset integration that preservers the biological variations between domain cluster types while removing batch effect across multiple tissues.

## BatchEval score

When it is difficult to measure batch effect for a dataset, BatchEval Pipeline provides a comprehensive assessment of the batch effect of aggregated data from different dimensions. BatchEval Pipeline is designed to calculate the final batch effect score by weighting the mean value with the formula as follows.

$$\text{BatchEval score} = \text{Mean}(\text{F1 score}_{LISI}, \text{F1 score}_{SS}, (1 - \text{DomainAcc})) \tag{10}$$

where DomainAcc is the accuracy of neural network domain classifier, Mean() is the mean function.

## DISCUSSION

Although many approaches have been developed to remove batch effect, there is still a lack of effective methods to evaluate batch effect for large-scale dataset integration, such as spatially resolved transcriptomics. The sources and effects of batch effect can vary greatly from one experiment to another. It is essential to analyze the most common potential sources of batch effect to improve the effectiveness of their removal and to facilitate the integration of data from different batches. Additionally, there is a need for an efficient method to accurately determine the extent to which data is affected by batch effect, and remove batch effect more precisely.

The BatchEval Pipeline simplifies batch pre-testing by offering a comprehensive evaluation report for data integration, making it particularly beneficial for multi-omics studies with multiple datasets or samples collected at different times. It provides statistical testing, batch effect metrics evaluation, and visualization, allowing researchers to efficiently explore and correct for batch effect in their data.

Although substantial progress has been made in identifying and evaluating batch effect, there is still room for improvement to enhance the accuracy and effectiveness of batch effect removal. The BatchEval Pipeline is a powerful tool for the evaluation of integrated large-scale gene expression datasets. It provides a quantitative measure of biological variance preservation and data integration mixing. Using the BatchEval Pipeline, users can objectively evaluate the presence and severity of batch effects in their integrated datasets. This feature makes the tool particularly valuable for researchers, who need to analyze large datasets, as it provides an easy and reliable way to assess data quality and ensures that downstream analyses are robust and reliable.

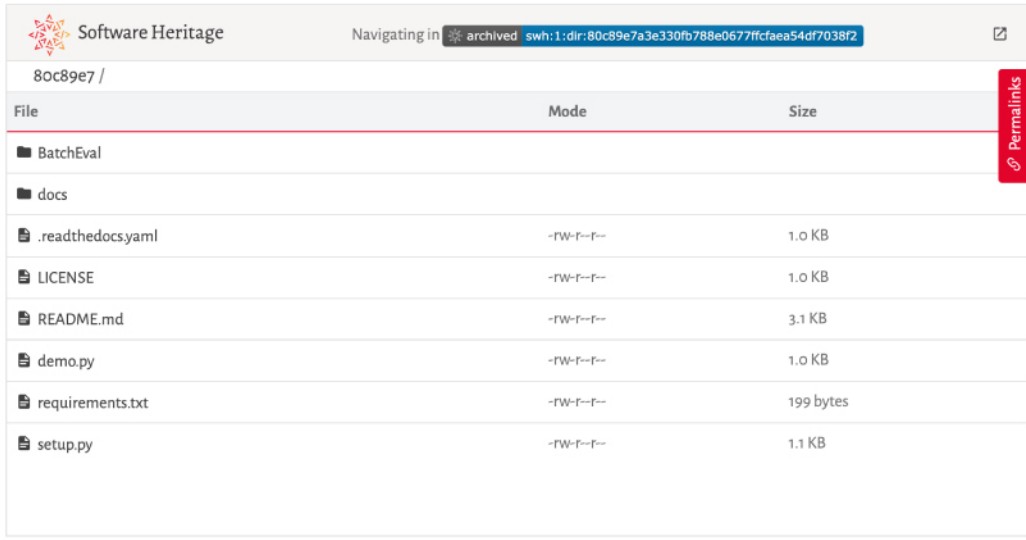

**Figure 4.** Software Heritage code snapshot.
https://archive.softwareheritage.org/browse/embed/swh:1:dir:80c89e7a3e330fb788e0677ffcfaea54df7038f2;
origin=https://github.com/STOmics/BatchEval;visit=swh:1:snp:585ccf2f25bd6a079f7c7bcbb08db7361554b300;
anchor=swh:1:rev:a908410fa1d855cc8c19cd26b5e57bd8b85449df/

## AVAILABILITY OF SOURCE CODE AND REQUIREMENTS

- Project name: BatchEval
- Project home page: https://github.com/STOmics/BatchEval
- Operating system(s): Platform independent
- Programming language: Python
- Tutorials: https://batcheval.readthedocs.io/en/latest/index.html
- License: MIT License
- RRID: SCR_024546

## DATA AVAILABILITY

The mouse olfactory bulb and embryonic brain datasets measured by Stereo-seq can be download from the MOSTA database [13], and the 10× Genomics Visium datasets can be download from 10× Genomics website [14]. Snapshots of the code and data are available in the GigaDB repository [21], and an archival code snapshot is available in Software Heritage (Figure 4) [22].

## DECLARATIONS

### Ethics approval and consent to participate
Not applicable.

### Competing interests
The authors are all employees of BGI Research.

## Authors' contributions

Conceptualization: CZ; Project administration and supervision: XX; Software: CZ; Data collection, processing, and application: ZC and HX; Project coordination: SF; Manuscript writing and figure generation: ZC and QK; Manuscript review: QK and ML.

## Funding

This work was supported by the National Key R&D Program of China (2022YFC3400400).

## Acknowledgements

We thank China National GeneBank for providing data support for this study.

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
