## [Reviewer Report]

Comments on revised manuscriptI find this paper to be much improved in this version. The authors have clearly worked hard to address my concerns and have addressed them in a satisfactory manner. I fully support the publication of this paper, and I believe their tools are a nice addition to the field.   W. Evan Johnson

---

## [Editor Report]

Editor’s AssessmentFor better data quality assessment of large spatial transcriptomics datasets this new BatchEval software has been developed as a batch effect evaluation tool. This generates a comprehensive report with assessment findings, including basic information of integrated datasets, a batch effect score, and recommended methods for removing batch effects. The report also includes evaluation details for the raw dataset and results from batch effect removal methods. Through peer review and clarification of a number of points it now looks convincing that this tool helps researchers identify and remove batch effects, ensuring reliable and meaningful insights from integrated datasets. Potentially making the tool valuable for researchers who need to analyze large datasets of this type, as it provides an easy and reliable way to assess data quality and ensures that downstream analyses are robust and reliable.

---

## [Reviewer Report]

Reviewer name and names of any other individual's who aided in reviewerChunquan LiDo you understand and agree to our policy of having open and named reviews, and having your review included with the published manuscript. (If no, please inform the editor that you cannot review this manuscript.)YesIs the language of sufficient quality?YesPlease add additional comments on language quality to clarify if neededIs there a clear statement of need explaining what problems the software is designed to solve and who the target audience is? YesAdditional CommentsIs the source code available, and has an appropriate Open Source Initiative license <a href="https://opensource.org/licenses" target="_blank">(https://opensource.org/licenses)</a> been assigned to the code?YesAdditional CommentsAs Open Source Software are there guidelines on how to contribute, report issues or seek support on the code?YesAdditional CommentsIs the code executable?YesAdditional CommentsIs installation/deployment sufficiently outlined in the paper and documentation, and does it proceed as outlined?YesAdditional CommentsIs the documentation provided clear and user friendly?YesAdditional CommentsAdditional CommentsIs there a clearly-stated list of dependencies, and is the core functionality of the software documented to a satisfactory level?YesAdditional CommentsHave any claims of performance been sufficiently tested and compared to other commonly-used packages? YesAdditional CommentsAdditional CommentsAre there (ideally real world) examples demonstrating use of the software? YesAdditional CommentsAdditional CommentsAny Additional Overall Comments to the Author1. Page 1, Lines 14-16. The authors indicate that “it is crucial to thoroughly investigate the batch effects in the dataset before integrating and processing the data”. The term “thoroughly” may be not accurate enough. The current method can alleviate the batch effects, but it can’t thoroughly solve the related problems. In addition, this work proposes a batch evaluation tool, such “reasonably evaluate the batch effects” may be more accurate than “thoroughly investigate the batch effects”. 2. In Figure 1, does the first box is “integrated datasets”? 3. Page 5, Line 168, and Page 6, Lines 169-175, the content of these two paragraphs is similar, with some redundant descriptions. It is recommended to organize and write them into one paragraph. 4. There is Table 1 in the table list, but Table 1 is missing in the main text. 5. Page 8, Discussion section, it is better to discuss the differences between the proposed tool and a similar tool “batchQC”, especially the advantages of the proposed tool. 6. Some other minor issues: Page 1, Line 22, “to do so” should be “to do it”. Page 3, Line 100, Ref. [13] should be cited when it first appears on Line 97. Page 4, Line 114 and Page 5, Line 146, “UMAP” should be given its full name when it first appears and abbreviated directly in the following text. The variable should be in italics, such as “p” on Page 4, Line 119, “H” on Page 6, Line 184.
RecommendationMinor Revisions

---

## [Reviewer Report]

Upload additional filesTRR-202306-01/form/BatchQC-Raw (1).pdfReviewer name and names of any other individual's who aided in reviewerW. Evan Johnson, Howard FanDo you understand and agree to our policy of having open and named reviews, and having your review included with the published manuscript. (If no, please inform the editor that you cannot review this manuscript.)YesIs the language of sufficient quality?YesPlease add additional comments on language quality to clarify if neededIs there a clear statement of need explaining what problems the software is designed to solve and who the target audience is? YesAdditional CommentsIs the source code available, and has an appropriate Open Source Initiative license <a href="https://opensource.org/licenses" target="_blank">(https://opensource.org/licenses)</a> been assigned to the code?YesAdditional CommentsHowever, the code could use substantial improvements As Open Source Software are there guidelines on how to contribute, report issues or seek support on the code?YesAdditional CommentsIs the code executable?YesAdditional CommentsIs installation/deployment sufficiently outlined in the paper and documentation, and does it proceed as outlined?NoAdditional CommentsThe manuscript is missing a section describing the software and its implementation.Is the documentation provided clear and user friendly?NoAdditional CommentsIs there enough clear information in the documentation to install, run and test this tool, including information on where to seek help if required?YesAdditional CommentsBut it took a while to get it installedIs there a clearly-stated list of dependencies, and is the core functionality of the software documented to a satisfactory level?YesAdditional CommentsHave any claims of performance been sufficiently tested and compared to other commonly-used packages? NoAdditional CommentsI think the most glaring deficiency in the paper is the lack of comparison with other methods. For example, there is no comparison of the tools available in BatchEval compared to other methods, such as BatchQC. Also, they mention that BatchQC might not work on larger datasets, but they perform no performance evaluation for BatchEval, and no comparison with BatchQC to demonstrate improved performance. Is test data available, either included with the submission or openly available via cited third party sources (e.g. accession numbers, data DOIs)?YesAdditional CommentsAre there (ideally real world) examples demonstrating use of the software? YesAdditional CommentsMissed opportunity--I think the most exciting thing I observed from the paper was that the example data were from spatial transcriptomics data! To my knowledge, existing batch effect methods are not directly adapted to manage these data (although they did mention tools like BatchQC cannot handle large datasets, which may be true). But they don’t mention anything about batch adjustment/evaluation in spatial data in the manuscript. I feel that if the authors address this niche it would increase the value/impact of their work! Is automated testing used or are there manual steps described so that the functionality of the software can be verified?YesAdditional CommentsAny Additional Overall Comments to the AuthorThis review was conducted and written by Evan Johnson, who developed the competing BatchQC software.   The authors provide an interesting toolkit for assessing batch effects in genomics data. The paper was clear and well-written, albeit I had a few concerns (see below). We were also able to download the associated software and test it out (comments below as well).   I think the most exciting thing I observed from the paper was that the example data were from spatial transcriptomics data! To my knowledge, existing batch effect methods are not directly adapted to manage these data (although they did mention tools like BatchQC cannot handle large datasets, which may be true). But they don’t mention anything about batch adjustment/evaluation in spatial data in the manuscript. I feel that if the authors address this niche it would increase the value/impact of their work!   In addition, this toolkit is written in Python, while BatchQC and other tools are written in R, so this is an advantage of the method as well—it addresses an audience that uses Python for gene expression analysis (not as big as the R community, but substantial). Their Python toolkit might also be more accessible to implementation in a pipeline workflow (for a core or large project) than R-based tools like BatchQC—this might be important to mention this as well.   I think the most glaring deficiency in the paper is the lack of comparison with other methods. For example, there is no comparison of the tools available in BatchEval compared to other methods, such as BatchQC. Also, they mention that BatchQC might not work on larger datasets, but they perform no performance evaluation for BatchEval, and no comparison with BatchQC to demonstrate improved performance.   Similarly, the authors claim: “Manimaran [10] has developed user-friendly software for evaluating batch effects. However, the software does not take into account nonlinear batch effects and may not be able to provide objective conclusions.” I don’t understand what the authors mean by “may not be able to provide objective conclusions” – BatchQC provides – several visual and numerical evaluations of batch effect – more so than even the proposed BatchEval does. Did the authors mean something else, maybe that the lack of non-linear correction may lead to less accurate conclusions?   A related concern: does BatchEval provide non-linear adjustments? I may have missed this, but it seems that BatchEval is not providing non-linear adjustments either. Also, regarding non-linear adjustments, the authors should show in an example the problems with a lack non-linear adjustments and show that pre-transforming the data before using BatchQC does not perform as well as the non-linear BatchEval adjustments.   In Equation 10, should “batchScore” be BatchEvalScore?   Also, in the bottom of Figure on page 15, should the “BatchQCScore” also be BatchEvalScore??  The manuscript is missing a section describing the software and its implementation.  I asked my research scientist, who recently graduated with his PhD in Bioinformatics, to assess the software and examples. First of all, much of the software is named “BatchQC”. I think this is confusing, since the method is really named BatchEval and it will be confused with BatchQC which is another existing/competing software.  Furthmore, it took him a significant effort to install the BatchEval software and get is working on our cluster. I would recommend the authors make their software more accessible and easier to install.   The output of the software was a nice .html report diagnosing the batch effects in the data—very useful (attached is a combined .pdfs of the .htmls that we generated). We were also able to generate a report for the harmony adjusted example using their code. One major disadvantage was that these reports are separate files, and this could get very complicated comparing cases using multiple batch effect methods that will all be in separate reports (refer to a recent single cell batch comparison that compared more than a dozen methods – Tran et al. Genome Biology, 2020 – it would be hard to use BatchEval for this comparison).   Also, it seems that the user is required to conduct the batch correction themselves, BatchEval does not help with the correction except for their example code for Harmony.   Finally, on comparing the raw and Harmony adjusted datasets, inspection of the visual assessments (e.g. PCA) show some improvement—although not a perfect correction. But must of the numerical assessments are still the sample. The BatchEvalScore in both cases leads to the conclusion “Need to do batch effect removal”. What’s missing is the difference or improvement that Harmony makes on its correction. Maybe this is just because Harmony doesn’t fully remove the batch effects? Or is there something not working in the code? Might be good to see another example where the batch effect correction improves the BatchEvalScore significantly.
RecommendationMajor Revisions